# The Use of Mobile-Based Ecological Momentary Assessment (mEMA) Methodology to Assess Dietary Intake, Food Consumption Behaviours and Context in Young People: A Systematic Review

**DOI:** 10.3390/healthcare10071329

**Published:** 2022-07-18

**Authors:** Brigitte Battaglia, Lydia Lee, Si Si Jia, Stephanie Ruth Partridge, Margaret Allman-Farinelli

**Affiliations:** 1Charles Perkins Centre, Sydney Nursing School, Faculty of Medicine and Health, The University of Sydney, Sydney, NSW 2006, Australia; bbat6930@uni.sydney.edu.au (B.B.); seoheelee.lydia@gmail.com (L.L.); margaret.allman-farinelli@sydney.edu.au (M.A.-F.); 2Engagement and Co-Design Research Hub, School of Health Sciences, Faculty of Medicine and Health, The University of Sydney, Sydney, NSW 2006, Australia; stephanie.partridge@sydney.edu.au; 3Prevention Research Collaboration, Sydney School of Public Health, Faculty of Medicine and Health, The University of Sydney, Sydney, NSW 2006, Australia

**Keywords:** m-health, mEMA, mobile ecological momentary assessment, dietary assessment method, young people, food consumption behaviour, eating behaviour

## Abstract

Mobile-based ecological momentary assessment (mEMA) offers a novel method for dietary assessment and may reduce recall bias and participant burden. This review evaluated mEMA methodology and the feasibility, acceptability and validity as a dietary assessment method in young people. Five databases were searched from January 2008 to September 2021 for studies including healthy young people aged 16–30 years and used mEMA for obtaining dietary intake data, food consumption behaviours and/or contextual factors. Data on the method used to administer mEMA, compliance with recording and validation were extracted. A total of 46 articles from 39 independent studies were included, demonstrating a wide variation in mEMA methods. Signal-contingent prompting (timed notification to record throughout the day) was used in 26 studies, 9 used event-contingent (food consumption triggered recordings), while 4 used both. Monitoring periods varied and most studies reported a compliance rate of 80% or more. Two studies found mEMA to be burdensome and six reported mEMA as easy to use. Most studies (31/39) reported using previously validated questions. mEMA appears to be a feasible and acceptable methodology to assess dietary intake and food consumption in near real time.

## 1. Introduction

In the transition from adolescence to young adulthood, young people experience significant personal development, increased independence and freedom of choice [1]. These major life transitions also present health challenges, including increased vulnerability to weight gain [1,2]. Compared to other age groups, young people have the highest mean gain in body mass index, placing them at a higher risk of overweight and obesity [3]. This weight trajectory is concerning as poor dietary behaviours and choices developed at a young age often persist into adulthood, affecting health outcomes later in life [1,2].

Identifying the behaviours and contextual factors that influence patterns of dietary intake in young people has been challenging to capture accurately due to the limitations of traditional dietary assessment methods [4]. For instance, most of the current methods used in practice, including 24 h recalls, food frequency questionaries and diet records, are subject to either recall and/or social desirability biases which reduce their validity [4]. Such self-report methods are often burdensome and subject to misreporting [5]. To overcome some of these limitations, assessment methods such as digital food diaries and image-based dietary assessments have emerged [4]. Further advances in technologies and their widespread societal adoption have created new opportunities to obtain and consider food consumption behaviours and the contextual factors surrounding eating events as they occur in everyday life [5,6].

Ecological momentary assessment (EMA) is a real-time data capture method originally used for psychological assessments that can monitor human phenomena as they occur in their natural environment [5,7]. EMA has appeared useful in obtaining social, psychological and environmental contexts surrounding dynamic patterns of diet behaviours simultaneously while removing the need for recall memory [6]. Most EMA research in recent times has been delivered over mobile technology due to its ubiquity, particularly with young people [8]. Mobile ecological momentary assessment (mEMA) has the advantage of being incorporated seamlessly into daily living by engaging the individual to provide samples of information in short bursts as it occurs in real time [9]. Sampling approaches to obtain information can be defined as signal-contingent or event-contingent. Signal-contingent sampling is a time-based approach that involves signalling the participant with a prompt to complete the mEMA (i.e., to recall the dietary intake or context that occurred within the recent time interval). Prompts can be sent at fixed times (intervals) or at random times throughout the day [5,10]. The other approach, event-contingent, involves the participant completing the mEMA when a relevant event has occurred (e.g., an eating or drinking event) [5]. For event-contingent sampling, this can be further differentiated into self-initiated or device-initiated. Self-initiated assessment requires the participant to self-initiate an mEMA recording in response to a specific event or behaviour of interest in which they engaged (e.g., eating). Device-initiated refers to the mobile device auto-initiating the mEMA in response to the detection of an event or behaviour (e.g., GPS tracking or wrist accelerometry) [10].

mEMA has potential to detect nutrition-related problems, allowing for earlier interventions in real-life settings. Previous reviews of EMA studies have focused on psychological and health-related behaviours such as emotions [11], alcohol use [12], craving and substance use [13], physical activity [10], sedentary activity [10] and dietary behaviours [14] across diverse age groups ranging from children and adolescents [15] to older adults [16]. Yet, few reviews have investigated the broad processes of mEMA to capture food consumption and related contextual factors of eating/drinking in young people. Additionally, young people are an understudied population with fast-changing priorities, high technology consumption and increased autonomy around eating and drinking, often leading to poor dietary choices; thus, it was deemed appropriate to target individuals aged 16–30 years [3,8]. Therefore, the current review aimed to close this knowledge gap by focusing on three key objectives: (1) examining the methodology of studies using mEMA to measure various aspects of food consumption, (2) evaluating the administration methods of mEMA and (3) assessing its feasibility, acceptability and validity as a dietary assessment method.

## 2. Materials and Methods

This systematic review followed the Preferred Reporting Items for Systematic Reviews and Meta-Analysis (PRISMA) checklist [17]. The review protocol is registered with the Open Science Framework (registration DOI: 10.17605/OSF.IO/WPC7Y; accessed on 24 January 2022).

### 2.1. Search Strategy

The following electronic databases were searched from 1 January 2008 to 8 September 2021: MEDLINE via Ovid, Embase via OvidSP, CINAHL via Ebsco, PsycINFO via Ovid and Scopus. The year 2008 was chosen because it marked the introduction of applications (apps) to the digital marketplace [18]. Key concepts (young people, mEMA and outcomes such as dietary intake, food consumption behaviours and contextual factors) and related terms were searched in all five databases using their appropriate syntax including truncations (*) and wildcards ($). The search strategy was limited to the English language and humans. The MEDLINE search strategy is presented in Table 1 and the full search strategies from the other four databases are available upon request.

### 2.2. Eligibility Criteria

The inclusion and exclusion criteria of the review were developed using a modified PICO framework based on population, EMA measurement method, setting and outcomes. The population was young people aged between 16 and 30 years old. This age range was identified as a time of growing independence and freedom of choice in young people before the typical life changes of marriage and children [1]. There was no restriction to participant characteristics such as gender, sex and race/ethnicity, except participants needed to be healthy without presenting chronic health conditions. The intervention criteria included any form of mEMA delivered on a portable electronic device. The primary outcomes included assessment of nutrition and diet such as dietary intake, food consumption behaviours and context if reported. The search strategy included all peer-reviewed primary research study designs conducted in humans. Qualitative studies and systematic reviews were excluded. All relevant studies from 2008 to present were included. There was no restriction placed on geographical location. The exclusion criteria included studies that were not peer reviewed, not in the English language and studies on young people with chronic health conditions such as eating and psychological disorders, diabetes, chronic dieting and post-bariatric surgery recipients, as these conditions do not represent typical food consumption behaviours [19]. Studies that solely focused on alcohol consumption were also excluded.

### 2.3. Screening and Selection of Studies

The identified studies obtained from all five databases were exported to the Endnote X9 citation management software [20] and then transferred to the Covidence platform [21]. Duplicate studies were screened and removed by automation tools firstly in Endnote and again in Covidence. Titles, abstracts and full texts were screened against the eligibility criteria. Studies that did not meet the inclusion criteria were excluded by following the hierarchy of the exclusion criteria and appropriate reasoning(s) provided. To reach a consensus, disagreements concerning eligibility were resolved through discussions between all review authors (BB, LL, MAF, SJ and SRP). The selection process of included studies was documented following the PRISMA flow diagram, presented in Figure 1.

### 2.4. Data Extraction

The data extraction table was designed and modified according to the two standardised forms—the Adapted STROBE Checklist for Reporting EMA studies ‘CREMAS’ used in previous reviews [5,10]. Two reviewers (BB and LL) independently extracted key information from all included studies. Data included: authors, year and country of publication, title, DOI, study design and overview, study aim(s), sample size, target population, participant characteristics and eligibility criteria. In addition, details of mEMA methodology were also extracted, including purpose/health domain, if mEMA training was provided, delivery mode, sampling approach, prompt frequency, prompt interval, reminders, prompt deactivation, the time required to complete one mEMA, monitoring period, latency, compliance rate, reactivity, missing data and incentives. Furthermore, attrition rate, user experience and participant burden, validity and outcomes of interest (such as dietary intake, food consumption behaviours and contextual factors) were extracted. Any discrepancies between the reviewers were resolved through discussion and consulting third experts (MAF, SP and SJ) when necessary.

### 2.5. Data Synthesis and Analysis

The study, sample characteristics and mEMA methodology of all included studies were summarised. The feasibility, acceptability and validity of each independent study was also summarised in tabular form. The findings were synthesised into a narrative review.

### 2.6. Quality Assessment

The Joanna Briggs Institute (JBI) critical appraisal checklist was used to assess the quality of each study according to its study design [22]. The included papers were either cross-sectional surveys or longitudinal cohort designs. Two authors (BB and LL) independently appraised each paper and together reached an overall agreement. Any discrepancies were discussed and resolved between all review authors (LL, BB, MAF, SP and SJ). The cross-sectional survey checklist had 8 items and the cohort 11 items. Each item was answered with either a ‘yes’, ‘no’, ‘unclear’ or ‘not applicable’. To determine the overall quality of evidence presented in the study, the following criteria were adapted from Shi et al. and used: ‘good’ (only ‘yes’ or ‘not applicable’ ratings), ‘fair’ (1 to 2 ‘no’ or ‘unclear’ ratings) and ‘poor’ (3 or more ‘no’ or ‘unclear’ ratings) [23].

## 3. Results

### 3.1. Study Selection

A total of 6615 records were identified from the five databases (Figure 1). After the automated and manual removal of duplicates, a total of 4203 records were screened. Then, another set of records (*n* = 3805) and duplicates (*n* = 59) were manually excluded, leaving a total of 339 full-text articles. Of these, 71 were abstracts only; thus, 268 full-text articles were assessed for eligibility, resulting in 45 studies meeting the inclusion criteria. An additional record was identified through forward chaining, which resulted in a total of 46 articles being included in this review. Data were extracted from each of the 46 articles and then merged with their respective studies where necessary (seven in total) and reported herein as independent studies (*n* = 39) [8,9,24,25,26,27,28,29,30,31,32,33,34,35,36,37,38,39,40,41,42,43,44,45,46,47,48,49,50,51,52,53,54,55,56,57,58,59,60,61,62,63,64,65,66,67].

### 3.2. Study and Sample Characteristics

Fourteen studies were conducted in the United States [8,24,25,26,27,28,29,30,33,35,36,40,43,45,50,59,63,64], seven in Germany [37,39,41,61,62,65,66,67], six in Austria and Germany [49,51,52,56], five in Australia [38,46,47,48,57,58], two in Austria [53,54] and one each in China [60], Poland [31,32], Sweden [42], Taiwan [44] and the Netherlands [9]. All studies were observational studies, including 3 cohort studies and 36 cross-sectional studies (Table 2). The studies were conducted with a range of target populations inclusive of young people such as adolescents, high-school, college or university students and adults including ethnic minority mothers [33] and healthcare workers [44]. The sample size of the studies ranged from 12 to 675 participants. Seven studies included females only [33,38,52,54,55,63,64]. Most studies had a higher female representation compared to male, ranging from 66% to 97%, and one study had a higher male population of 72% [30]. Studies that sampled from a general population reported a mix of races and ethnicities, including Caucasian (23% to 83%), Asian (4% to 42%), Hispanic (3% to 29%) and African American (1% to 23%). One study sampled adolescents with lower socioeconomic status with 67% of participants identifying as Hispanic [27,28,29]. The mean BMI or percentile was reported in 27 studies [26,27,28,29,31,32,33,35,37,38,39,40,41,42,44,49,51,52,53,54,55,56,57,58,59,60,61,62,64,65,67]. All but two [31,32,33] studies had a majority of participants within the healthy BMI range of 20 to 24.9 kg/m^2^. Most studies sampled students completing a tertiary level of education with 27% to 75% employed either part or full time.

### 3.3. The Use of mEMA for Dietary Assessment Method and Diet-Related Behaviours

Among the 39 mEMA studies, 25 reported dietary intake data [9,24,26,27,30,31,32,33,35,36,39,41,43,44,45,46,48,50,52,53,54,57,58,61,62,63,65,66]. Of these studies, 14 examined intake of a variety of food, food groups and beverages [9,24,27,30,33,41,43,45,48,50,52,57,58,61,62,63], 2 assessed food intake only [65,66], 7 specifically focused on the intake of high energy foods and beverages [26,35,36,39,44,53,54], 1 study examined intake of high-fat foods [31,32] and 1 assessed sugar-sweetened beverage consumption only [46,47]. In addition to dietary intake, 34 of the 39 studies examined associated food consumption behaviours [9,24,26,27,30,31,32,33,35,36,38,39,40,41,42,43,44,45,46,48,49,50,51,52,54,55,56,57,58,59,61,62,63,64,65,66,67]: these included comfort eating and overeating behaviours [38,55], healthy versus unhealthy eating [56], and eating to match their dieting goals [52], food desires [37,51], thoughts and cravings of snacks and principal meals [53], and emotions and feelings [60]. Other contextual factors have been explored in 37 of the 39 studies [9,24,26,27,30,33,35,36,37,38,39,40,41,42,43,44,45,46,48,49,50,51,52,53,54,55,56,57,58,61,62,63,64,65,66,67]. There were 8 assessed environmental factors (e.g., location of consumption or work schedule), 11 assessed social factors (e.g., whom participants were with during consumption), 24 examined psychological factors (e.g., stress, affect, mood), 15 examined biological factors (e.g., hunger, cravings, appetite) and five studied physical factors (e.g., availability of foods). Some studies investigated other lifestyle-related chronic disease risk factors such as physical activity, sedentary activity, cigarette craving and reasons for smoking and sleep.

In most studies, mEMA was used as the sole method for collecting dietary information. Eleven studies reported using mEMA in combination with other technologies to collect data on health outcomes. One study used mEMA concurrently with a chewing sensor and digital food scale as part of the development of a ‘mobile- or mHealth system’ to evaluate the acceptability, usability and limitations of this system to self-monitor dietary habits [42]. Two studies combined mEMA with an accelerometer to assess the interrelations of physical activity and dietary intake [45,60] and one with a smartphone sensor [48] to collect additional data on physical activity and sedentary behaviour passively. Two studies collected saliva samples with each mEMA data entry to assess physiological stress markers: cortisol, alpha-amylase and flow rate [61,62]. One study added the additional function of photographic food records which were self-initiated by the participant separate from the mEMA items [33]. Four of the applications used for mEMA had geographical location (GPS) or information systems (GIS) to collect information on food environments and daily activity [25,33,48,60]. One study directly compared mEMA to handwritten EMA to ascertain differences in compliance [26].

### 3.4. Data Input Modalities

Table 2 presents the data input modalities employed by the studies. A smartphone application was the most common delivery mode used by 26 out of 39 studies [8,9,24,25,30,31,32,33,35,37,41,42,45,46,47,48,49,51,52,53,54,56,57,58,59,60,65,66,67]. Of the remaining studies, six used short text message service (SMS) [26,38,39,40,50,55], three used a personal digital assistant (PDA) [27,28,29,43,64], one used email [36], one used an iPod Touch [61,62] and another used a palmtop computer [63]. One study gave participants the option to be sent the EMA survey via email or SMS with a link to the survey [44]. The smartphone applications selected by researchers varied among studies with only four using the same application (MovisensXS^®^) [41,45,65,66]. The other applications were unique to each study. The SMS delivery modes utilised automated bulk messaging systems to send a hyperlink to the EMA survey at each prompt [38,39,40,55]. One study did not report the automated bulk text messaging service used but specified using SurveySignal^®^ for the EMA survey [50]. One survey used SMS to prompt via mProve^®^ software and required participants to directly send a text reply with their responses [26].

### 3.5. mEMA Sampling Approach, Prompt Frequency and Interval

Both signal-contingent and event-contingent sampling approaches were used in the included studies (Table 2). For the event-contingent sampling, it was further classified to whether it was self-initiated, device-initiated or both, as proposed in the CREMAS framework [5]. Of the 39 studies, 27 used signal-contingent sampling [8,9,24,25,26,31,32,33,34,35,36,37,38,39,40,44,45,48,49,50,51,52,53,54,55,56,60,61,62,63,64] and 7 studies [41,43,46,47,59,65,66,67] implemented an event-contingent, self-initiated sampling design. Of the seven studies, participants were asked to report eating or drinking occasions either before [66], during [41,42,46,47,59,65,67] or soon after the occasion [43]. One of the studies allowed forgotten meals to be recorded afterward, and these were identified as such for the analysis [65]. Only one study used event-contingent sampling that utilised both device-initiated and self-initiated forms [42]. Four studies incorporated a sampling design that was both signal-contingent and event-contingent via self-initiation [27,28,29,51,57,59]. Only 2 of the 39 studies included an end of the day assessment for participants to complete [27,28,29,33].

The frequency and schedule of prompts were reported in all 32 signal-contingent studies [8,9,24,25,26,27,28,29,30,31,32,33,34,35,36,37,38,39,40,42,44,45,48,49,50,51,52,53,54,55,56,57,58,60,61,62,63,64]. The frequency of prompts varied from once daily [36,48] to 24 times per day [64]. Most studies (59%) sent prompts between 4 and 7 times a day [9,26,27,28,29,30,31,32,33,34,37,38,39,40,44,45,49,50,51,52,53,54,55,56,57,58,60,61,62,63] (Table 3). All but 1 of the 32 signal-contingent studies reported a prompting schedule which was either at a fixed (*n* = 19) [9,26,31,32,33,35,36,39,40,49,51,52,53,54,56,61,62,64] or random time (*n* = 12) [8,24,25,27,28,29,30,37,38,44,45,48,50,55,57,58,60]. The fixed times included every hour [35,64], every two and a half hours [40,51,52,56], every three hours [9,33,34,39,51,53,54,61,62], every four hours [31,32,49,52] and once at night [36]. Some studies sent prompts at randomised times within fixed time windows [24,27,37,44,48]. For example, prompts were sent twice between the four-time windows: 9 a.m.–12 p.m., 12–3 p.m., 3–7 p.m. and 7–10 p.m. [8,24,25]. In other studies, prompts were sent at random times within a single, wider time window, making sure they were sent at least one to three hours apart [30,38,45,50,55]. For example, in one study, prompts were sent between 10 a.m. and 8 p.m. with a minimum of 1 h apart [55]. Only two studies did not report a prompt schedule [57,58,60]. In 4 of the 39 studies, the prompt schedule was based the individual’s typical predetermined meal times and bedtime [26], adjusted according to school and non-school days [27,28,29], individually selected to cover mornings, middays and late afternoons [33,34] and accustomed to shift work schedule and wake–sleep pattern [44].

### 3.6. Monitoring Period

Across the 39 mEMA studies, total monitoring periods varied from two days to nine months (Table 3). Most studies monitored for a period between 2 and 14 consecutive days [9,26,27,28,29,30,31,32,35,36,37,38,39,40,41,43,44,45,49,50,51,52,53,54,55,56,57,58,59,60,61,62,63,64,65,66,67]. Only one study monitored for three or four non-consecutive days, including one weekend day over a two-week duration [46,47], and one study monitored two blocks of six days over two weeks [51]. Two studies had varied monitoring periods such as 14 to 21 days [42] and a 16-week schedule of follow-up EMAs over 22 weeks [48]. Comulada et al. monitored for six months [33].

### 3.7. Protocol Adherence (Training, Reminders, Deactivation)

Training was provided to participants prior to commencing the EMA protocol in 30 out of 39 studies [8,24,25,26,27,28,29,31,32,33,35,37,40,41,42,43,44,45,46,47,49,51,52,53,54,56,57,58,60,61,62,63,64,65,66]. Around half of the included studies [8,24,25,27,28,29,30,33,35,36,37,38,39,40,42,43,48,51,54,55,59,61,62,64,66,67] 21 out of 39 sent no prompts for completion. Timing and frequency of reminders differed across studies. Reminders were sent either once or twice at intervals ranging from 5 to 60 min [9,31,32,44,45,50,52,53,54,56,63]. Two studies sent a morning and an evening reminder [41,65] and four studies reminded their participants once a day to enhance compliance [26,49,57,58,60]. For one study using event-contingent (self-initiated) mEMA sampling, a reminder was sent if three hours elapsed without an eating occasion reported [46,47]. Prompts were reported as being deactivated by the device in 13 studies [8,24,25,31,32,37,40,44,45,49,50,51,52,56] or by the user in one study if not responded to in real time [63]. Prompt deactivation times ranged from within 15 min up to 1 h [37,45] [8,24,25,31,32,40,44,49,50,51,52,56]. Three studies allowed the user to delay responding to a prompt for up to one hour when impossible or unsafe to reply [51,52]. Only 6 out of 39 studies reported the average time taken for the participant to receive and respond to prompts, known as latency [8,24,25,26,36,49,51]. The latencies in these studies ranged from 7.25 min [8,24,25] to up to 6 h [36]. Six studies reported the time taken to complete one mEMA survey [8,9,24,25,27,28,29,36,45,64]. The times ranged from 1 min [8,24,25] to 10 min [36], with less than 5 min being the most common.

### 3.8. Feasibility

Compliance was the key feature for assessing the feasibility of mEMA. A total of 30 of the 39 studies reported compliance and this ranged from less than 1% to 98% [9,24,26,27,28,29,30,31,32,33,35,36,37,39,40,42,44,45,46,48,49,50,51,52,53,54,56,57,58,59,61,62,63] (Table 3). Twenty studies recorded a compliance rate over 80% [26,27,28,29,31,32,35,36,37,39,40,45,46,47,49,51,52,53,54,56,57,58,61,62]. Three studies found that the response rates were less likely to be answered on weekends than weekdays [8,24,25,30,33,34], and one study found compliance increased from 70% at week one to 76% in week two [42]. A total of 8 of the 39 studies stated whether the compliance rate was associated with other variables [9,24,30,31,32,45,55,61,62,63]. Of these, gender was significantly associated with the mEMA completion, with males less likely to complete them. Four studies reported a compliance rate below 60% [44,48,50,59] with one of these studies being an extreme outlier of <1% [48].

### 3.9. Acceptability

Acceptability was assessed by missing data, participant burden, reactivity and usability. In 17 of the 39 studies, missing data ranged from 0% to 31% with reasons being technical issues, incompleteness, invalid reporting and low response rate [8,30,31,32,33,36,37,38,39,40,41,44,45,55,56,57,58,61,62,66]. In the 10 studies that reported participant burden, they stated mEMA reduced burden due to brevity, ability to skip prompts and focusing on types and not portion sizes of food consumption [24,33,35,39,41,43,45,52,57,58]. Two studies reported high burden was due to the duration of the mEMA protocol (14 days [44] and up to 22 weeks [48]). Only seven studies reported on reactivity, that is, if the mEMA altered usual behaviour [35,39,49,53,55,56,64]. Six found no reactivity associations between eating behaviours and snack-related thoughts while using mEMA [35,39,53,55,56,64]. One study found low/moderate reactivity [49]. Similarly, most studies did not report on the usability of mEMA, with only six reporting data [9,26,33,42,43,59]. They highlighted that the app was easy, satisfying and comfortable to use; however, some stated it was slow, the colour scheme was not appealing [42], they experienced technical difficulties [59] and the surveys were time-consuming [9].

### 3.10. Validity of mEMA Methodology

In 4 of the 39 studies, the mEMA methodology was pilot tested [8,29,31,34,47]. Three of these studies included validation as a dietary assessment method by comparing it to standard tools [8,31,34,47]. The validation of one study occurred using the online Automated Self-Administered 24-Hour (ASA24) dietary recall and found high match rates across food types (79 to 94%) [8]. Another study used a three-day estimated food record [31] and one validated with food frequency questionnaires, resulting in high inter-method reliability (*p* < 0.05) [34]. Lastly, one study validated their mEMA against measured energy expenditure using a validated multi sensor monitor (SenseWear Armband), comparable in accuracy and reliability to doubly labelled water [47]. The study found good agreement and reliability after comparing the estimated energy intake obtained from the mEMA [47]. The validity of the mEMA items used was reported in 31 out of 39 studies. Dietary intake and food consumption behaviour items included in 17 studies were based on previous research [8,9,24,25,27,28,29,30,35,36,37,39,40,41,45,49,50,56,57,58,61,62,65]. Most studies reported using validated questionnaires of relevance to their study aims. Questionnaires were used to collect dietary intake data [31,36] and food consumption behaviours [56,67], including food addiction and craving [51,52]. In 3 of the 31 studies, country specific databases and surveys were used [44,48,66]. Nine studies reported data on emotion, a psychological contextual factor, by using questionnaires in their mEMA items [27,28,29,35,38,40,44,49,55,56]. To collect data on the specific emotion of stress, questionnaires were used by six studies [27,28,29,36,48,49,52,56].

### 3.11. Quality Assessment

Among the 36 independent studies that used cross-sectional designs, 16 were assessed as fair quality [27,28,29,30,31,32,33,38,39,44,45,46,52,53,56,57,58,61,62,63], 19 were poor [9,26,36,40,41,42,43,49,50,51,54,55,59,60,64,65,66,67] and only 1 study was rated as good quality [35] (Table 4). For the poor-quality studies, it was hard to determine the inclusion criteria, exposure/outcome measures, confounding factors and type of statistical analysis used. Of three longitudinal cohort studies, two were of fair quality [24,48] and one was rated as poor quality [37]. Regarding the fair-quality studies, the strategies for dealing with the confounders were unclear or not stated, and follow-up information was not addressed. The poor-quality study did not include eligibility criteria and further details on follow-up were not explained clearly.

Cross-sectional questions (1–8): 1. Were the criteria for inclusion in the sample clearly defined? 2. Were the study subjects and the setting described in detail? 3. Was the exposure measured in a valid and reliable way? 4. Were objective, standard criteria used for measurement of the condition? 5. Were confounding factors identified? 6. Were strategies to deal with confounding factors stated? 7. Were the outcomes measured in a valid and reliable way? 8. Was appropriate statistical analysis used?

Cohort questions (1–11): 1. Were the two groups similar and recruited from the same population? 2. Were the exposures measured similarly to assign people to both exposed and unexposed groups? 3. Was the exposure measured in a valid and reliable way? 4. Were confounding factors identified? 5. Were strategies to deal with confounding factors stated? 6. Were the groups/participants free of the outcome at the start of the study (or at the moment of exposure)? 7. Were the outcomes measured in a valid and reliable way? 8. Was the follow up time reported and sufficient to be long enough for outcomes to occur? 9. Was follow up complete, and if not, were the reasons for loss to follow up described and explored? 10. Were strategies to address incomplete follow up utilized? 11. Was appropriate statistical analysis?

## 4. Discussion

This review shows that mEMA can be used to collect data on food and beverage intake, dietary habits, food behaviours and contexts such as the physical environment, emotions and social interactions in young adult populations. Most studies employ smartphone applications to deliver signal-contingent prompts and collect data from participants rather than event-triggered prompts, perhaps because of the need for memory to trigger recording. Compliance was above 80% in half the studies with varying schedules of prompts, reminders and frequency and duration of data collection. Attrition from studies ranged from almost none to one in two participants, but overall attrition and compliance would appear to support the feasibility and acceptability of mEMA. A major limitation of the current evidence base is that the quality of studies is overall poor, with only one rated as good quality, and most assessment of dietary intake has not been validated. Thus, further studies may be needed to clarify findings and determine the most effective protocols to administer EMA in young adults maximising compliance and participation.

Factors that would improve compliance might include training in the system, lower participant burden and shorter study durations. More than half of the studies reported training in advance of the study and Stone and Shiffman have previously made recommendations that encourage reporting on training status when documenting EMA protocols in methods [68]. The study with the lowest compliance of <1% monitored for 16 weeks once per day over a 22-week period. However, other studies ran for longer times with better compliance. It remains unclear if event-contingent (self-initiated) or signal-contingent prompts lead to better recording. The former depends on participants’ memory to record in response to food ingestion, which may be problematic but may also mean improved accuracy as it is in real time. Conversely, sending prompts to record may occur more distal to food consumption and hence rely on memory to recall food and beverage intake. With signal-contingent sampling, the frequency of prompts and programming of prompts around reported mealtimes reduces the time period between ingestion and recording, thereby eliminating the need to recall what was eaten over a longer period of time (i.e., 24 h, 3 days) [4]. It should be noted only one study used event-contingent sampling via the use of a wearable device that detected eating. A previous review of children and adolescents reported no advantage of wearables over mobile EMA only [69]. However, as a recent review noted there are few wearable sensors available that could be used for event-contingent mEMA.

Finding the appropriate number and duration of sampling to maintain compliance to the protocol remains a challenge for many researchers. In the current review of young adults, it was found among the eight studies with the highest compliance rates (>90%) [26,37,39,54,57,58,61,62] that five signal-contingent collections of data a day was the most common frequency employed. A previous review and meta-analysis of EMA in children and adolescents reported that the average daily number of prompts was 4.2 for non-clinical participants and 3.6 for clinical participants, and the weighted average compliance rate was 78.3%. The duration of the sampling period did not alter compliance [69]. In a meta-analysis by Williams et al., 68 data sets (41 non-clinical and 27 clinical) in adults were included and it was estimated that overall compliance to mEMA was 81.9% [70]. The median number of prompts per day was found to be five in non-clinical data sets and four in clinical data sets. Interestingly, less frequent prompting of 1–3 prompts per day increased compliance in non-clinical participants compared to 6 or more with 87% and 79.4% compliance rates, respectively. No significance was found in clinical data sets. The meta-analysis by Williams et al. has given insight into compliance relating to prompt frequency; however, the focus of their review was not solely on diet but rather health-related behaviours, which included a small proportion of studies on eating behaviours [70]. Schembre et al. conducted a systematic review on mEMA focusing on diet studies inclusive of both children and adults. They found prompt frequencies of the included signal-contingent studies ranged from 3 to 14 prompts per day with a mean response rate of 79% [4]. Similarly, Maugeri and Barchitta conducted a review in children and adults and found the prompt frequency ranged from 1 to 14 times per day [14].

There were only four studies that performed validation studies for the use of mEMA as a dietary assessment method by comparison to traditional methods such as food records, 24 h recalls and a novel measure of energy expenditure instead of the traditional doubly labelled water. Hence, this limits the ability to recommend mEMA as an effective dietary assessment method. However, there are other advantages of the mEMA method in studying food consumption in that it allows real time evaluation of the social, emotional and environmental context in a way a 24 h recall cannot. Thus, the decision of whether to employ mEMA should depend on the motive for monitoring diet. Clearly mEMA would not appear to be the method of choice for large epidemiological cohort studies of diet disease relationships as outlined in a previous systemic review [4]. However, it may well be useful to monitor changes in nutrition behaviours in different contexts and in response to an intervention or to plan an intervention in an individual or population. Collection of food and beverage consumption to yield macronutrient and micronutrient data is not required in many scenarios in order to improve an individual’s dietary behaviour.

Overall, as noted by others [5,71] assessing the mEMA methodology and its feasibility, acceptability and validity remained challenging in this review due to the inconsistencies and absence of key factors in reporting across studies. The major strengths of the current review include an extensive search of multiple databases, resulting in an ample number of mEMA studies for screening and selection. The review focused on mEMA, which is applicable to the digital-savvy 16- to 30-year-olds. A strength of this review is compliance with recognised standards for reporting in systematic reviews and EMA studies and the quality appraisal [5,17]. However, it is acknowledged that this systematic review is not without some limitations. The year 2008 was selected as the year applications were introduced to the digital marketplace and accessible on smartphones, but this may mean some EMA studies with personal digital assistants and text messages were omitted. However, our purpose was to provide an evidence base for those wishing to employ mEMA in the current digital environment. In addition, the target population was young people, so the findings cannot be extrapolated to other age groups such as early adolescents and older people. Only English studies were included, yielding a language bias as per the eligibility criteria.

## 5. Conclusions

The current review of 39 independent studies offers unique insights into the uses of mEMA in young people aged 16 to 30 years old. This population is well documented to have poor diet quality and experience the most weight gain among adults. Measuring their food consumption and the context of this consumption is an important step in formulating intervention. mEMA has demonstrated potential to become a feasible and acceptable methodology in assessing food and beverage consumption with the advantage of providing social, emotional, and food environment contextual information. Further research in the technology of wearables and detection of eating as well as validated questionnaires for data collection would advance the field.

## Figures and Tables

**Figure 1 healthcare-10-01329-f001:**
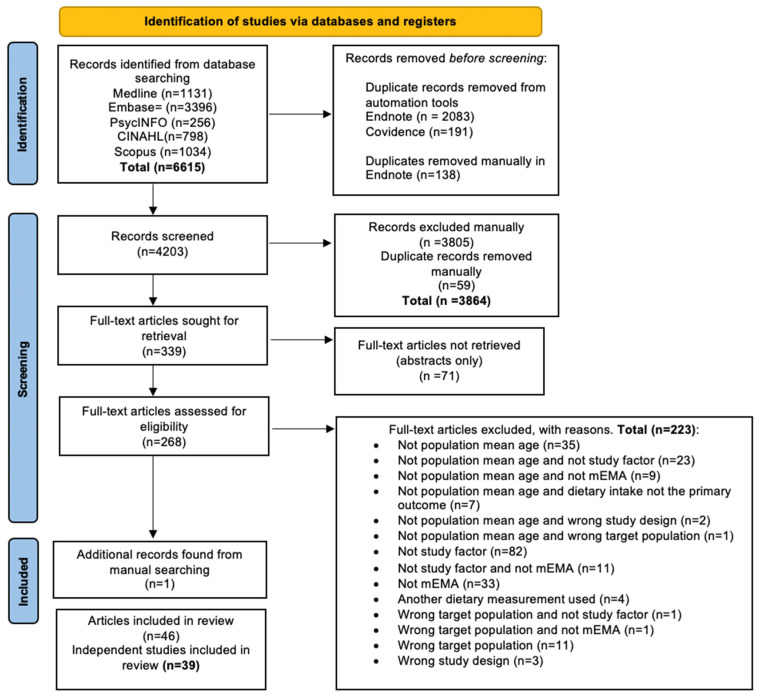
The Preferred Reporting in Systematic Review and Meta-Analyses (PRISMA) flow diagram documenting the literature search process [17].

**Table 1 healthcare-10-01329-t001:** A sample search strategy from MEDLINE.

#	Query	Results from 8 September 2021
1	Adolescent/	2,120,027
2	Young Adult/	944,129
3	Adult/	5,231,956
4	Adolescen *.tw.	303,187
5	Teen *.tw.	32,133
6	Youth *.tw.	82,599
7	Adult *.tw.	1,361,434
8	Emerging adult *.tw.	2964
9	(Young adj2 (Adult * or person * or people * or wom#n or m#n or female * or male * or boy * or girl *)).tw.	230,385
10	1 or 2 or 3 or 4 or 5 or 6 or 7 or 8 or 9	6,869,361
11	Ecological Momentary Assessment/	938
12	Mobile Applications/	8538
13	Digital Technology/	268
14	exp Computers, Handheld/	10,256
15	EMA.tw.	9726
16	mEMA.tw.	82
17	ecological momentary assessment *.tw.	2193
18	ecological momentary intervention.tw.	53
19	mobile ecological momentary assessment *.tw.	17
20	mobile-based ecological momentary assessment *.tw.	5
21	ambulatory assessment *.tw.	396
22	experience sampl *.tw.	1406
23	real-time data.tw.	1825
24	((food or diet) adj2 tracking).tw.	122
25	((electronic or daily) adj1 diar *).tw.	3463
26	personal digital assistant *.tw.	1012
27	((repeat * or real-time) adj2 sampling).tw.	2049
28	(within adj1 (person or subject *)).tw.	18,788
29	(between adj1 (person or subject *)).tw.	21,215
30	mobile health technolog *.tw.	482
31	((mobile or smart or cell *) adj1 (device * or phone *)).tw.	18,619
32	11 or 12 or 13 or 14 or 15 or 16 or 17 or 18 or 19 or 20 or 21 or 22 or 23 or 24 or 25 or 26 or 27 or 28 or 29 or 30 or 31	88,291
33	Nutrition Assessment/	16,225
34	exp Nutrition Surveys/	28,923
35	Diet Records/	5873
36	exp Beverages/	148,276
37	exp Food/	1,355,200
38	Nutrients/	3974
39	exp Meals/	7032
40	((diet * or food * or nutr *) adj3 (knowledge or history or assessment * or record* or recall * or analysis or survey *)).tw.	71,520
41	(Nutrition * adj3 (chang * or intake * or quality or status or maintain * or maintenance or poor)).tw.	56,659
42	((Intake * or consum *) adj3 (food * or drink * or beverage * or diet * or energy or nutrient *)).tw.	190,164
43	portion size *.tw.	1740
44	serving size *.tw.	528
45	33 or 34 or 35 or 36 or 37 or 38 or 39 or 40 or 41 or 42 or 43 or 44	1,603,381
46	Feeding Behavior/	87,461
47	Health Behavior/	53,419
48	Drinking Behavior/	6795
49	Eating/	55,529
50	Drinking/	14,497
51	exp Life Style/	101,509
52	Food Preferences/	15,330
53	Diet, Reducing/	11,306
54	Snack *.tw.	8719
55	(life style or lifestyle).tw.	112,229
56	food consumption behavio?r *.tw.	111
57	nutr * behavio?r *.tw.	1102
58	eat *.tw.	107,019
59	drink *.tw.	149,261
60	(Meal * adj3 (skip * or miss * or pattern * or tim *)).tw.	4663
61	(Eat * adj3 (habit * or pattern * or behavio?r *)).tw.	22,876
62	(Food * adj3 (content or habit * or quality or choice *)).tw.	21,264
63	(Diet * adj3 (habit * or pattern * or practice * or chang * or quality or behavio?r *)).tw.	57,954
64	46 or 47 or 48 or 49 or 50 or 51 or 52 or 53 or 54 or 55 or 56 or 57 or 58 or 59 or 60 or 61 or 62 or 63	621,175
65	Work/	20,184
66	Schools/	43,223
67	Universities/	45,308
68	Restaurants/	4129
69	supermarkets/	123
70	exp Marketing/	36,116
71	Social Media/	11,151
72	exp Mass Media/	46,714
73	Workplace/	25,661
74	Fast foods/	2490
75	Food dispensers, automatic/	355
76	Weather/	10,893
77	Income/	31,348
78	convenience.tw.	44,499
79	home.tw.	242,032
80	fast food outlet *.tw.	255
81	supermarket *.tw.	3978
82	((grocery or convenience) adj1 store *).tw.	2048
83	(food adj1 (court * or outlet * or price *)).tw.	1409
84	(fast food * or fastfood *).tw.	3888
85	restaurant *.tw.	6136
86	cafe *.tw.	5430
87	(take away or takeaway).tw.	774
88	(take out or takeout).tw.	399
89	(pub or pubs).tw.	2634
90	public bar *.tw.	33
91	(club or clubs).tw.	15,181
92	(work place * or workplace *).tw.	47,820
93	vending machine *.tw.	644
94	street food *.tw.	229
95	((food * or eat *) adj3 (away or out or outside)).tw.	2746
96	cost of food.tw.	260
97	time constraint *.tw.	4603
98	(food adj3 market *).tw.	2071
99	65 or 66 or 67 or 68 or 69 or 70 or 71 or 72 or 73 or 74 or 75 or 76 or 77 or 78 or 79 or 80 or 81 or 82 or 83 or 84 or 85 or 86 or 87 or 88 or 89 or 90 or 91 or 92 or 93 or 94 or 95 or 96 or 97 or 98	611,229
100	exp Peer Group/	22,672
101	Friends/	5806
102	Family/	79,729
103	Parents/	69,812
104	Siblings/	12,134
105	Culture/	33,720
106	Religion/	15,004
107	Unemployment/	7352
108	Employment/	47,868
109	cultural diversity/	12,115
110	colleague *.tw.	36,918
111	peer pressure *.tw.	1285
112	(social adj3 (value * or desirabilit * or norm * or interaction * or support or setting * or context *)).tw.	99,210
113	(ethnic * adj1 (group * or value *)).tw.	35,437
114	100 or 101 or 102 or 103 or 104 or 105 or 106 or 107 or 108 or 109 or 110 or 111 or 112 or 113	443,731
115	exp Emotions/	267,374
116	Attitude/	49,588
117	Cognition/	108,153
118	exp Stress, Psychological/	141,385
119	exp Body Image/	18,464
120	mood.tw.	79,081
121	affect regulation.tw.	1169
122	belief *.tw.	90,267
123	(self adj1 (control or regulation or esteem)).tw.	37,252
124	stigma.tw.	27,112
125	115 or 116 or 117 or 118 or 119 or 120 or 121 or 122 or 123 or 124	715,144
126	Hunger/	5674
127	Satiety Response/	2538
128	Smell/	16,755
129	Taste/	24,034
130	Vision, Ocular/	25,841
131	Hormones/	37,194
132	Craving/	1789
133	Thirst/	3306
134	Appetite/	8199
135	texture.tw.	32,181
136	sensory.tw.	191,635
137	flavo?r.tw.	16,962
138	palatab *.tw.	7764
139	taste sensitivit *.tw.	821
140	visual * appeal *.tw.	285
141	126 or 127 or 128 or 129 or 130 or 131 or 132 or 133 or 134 or 135 or 136 or 137 or 138 or 139 or 140	344,364
142	Food Security/	189
143	Food Insecurity/	462
144	food availabilit*.tw.	4141
145	food accessibilit *.tw.	110
146	142 or 143 or 144 or 145	4859
147	64 or 99 or 114 or 125 or 141 or 146	2,391,555
148	10 and 32 and 45 and 147	1589
149	limit 148 to (English language and humans and yr = “2008–Current”)	1129

**Table 2 healthcare-10-01329-t002:** Summary of methodological features of mobile-based ecological momentary assessment (mEMA) studies in young people. The Table was modified from Liao et al. ‘Adapted STROBE Checklist for Reporting EMA Studies (CREMAS) [5]’ and ‘Table 1 Data Extraction Scheme’ from Degroote et al. [10].

**Study Design**	Cross-Sectional Study *n* (%)	37 (95%)
	Cohort study *n* (%)	3 (5%)
**EMA sampling approach**	Signal-contingent *n* (%)	27 (69.25%)
	Event-contingent (self-initiated) *n* (%)	7 (18%)
	Event-contingent (device-initiated) *n* (%)	0
	Event-contingent (self-initiated and device-initiated) *n* (%)	1 (2.5%)
	Signal-contingent and event-contingent (self-initiated) *n* (%)	4 (10.25%)
**Delivery mode**	Smartphone app *n* (%)	26 (67%)
	Personal digital assistant *n* (%)	3 (8%)
	Palmtop computer *n* (%)	1 (2.5%)
	iPod Touch *n* (%)	1 (2.5%)
	Email	1 (2.5%)
	Automated SMS link to online survey *n* (%)	6 (15%)
	Email or automated SMS link to online survey *n* (%)	1 (2.5%)
**Monitoring period**	< 4 days *n* (%)	2 (5%)
	4–7 days *n* (%)	20 (51%)
	8–14 days *n* (%)	13 (33%)
	15–21 days *n* (%)	1 (2.5%)
	>1 month *n* (%)	3 (7.7%)
**Prompt frequency (times/day)**	<4 *n* (%)	2 (5%)
	4–6 *n* (%)	23 (59%)
	>6 *n* (%)	6 (15%)
	N/A *n (%)*	8 (21%)
**Prompt interval**	Fixed *n* (%)	18 (46%)
	Random *n* (%)	13 (33%)
	N/A *n* (%)	8 (21%)
**Reminders sent**	Yes *n* (%)	18 (46%)
	Not reported *n (%)*	21 (54%)

**Table 3 healthcare-10-01329-t003:** Feasibility of mobile-based ecological momentary assessment (mEMA) for signal-contingent and event-contingent sampling approaches in young people. The table was modified from Liao et al. ‘Adapted STROBE Checklist for Reporting EMA Studies (CREMAS)’ [5] and ‘Table 1 Data Extraction Scheme’ from Degroote et al. [10].

First Author, Publication Year, Country	EMA Sampling Approach	Monitoring Period, Prompt Frequency (Per Day), Prompt Interval	Feasibility	
			Response Rate	Factors Influencing Response Rate
Ashurst et al., 2018 [24], Bruening et al., 2016 [25], Bruening et al., 2016, USA [8]	Signal-contingent	4 waves, 4 days each, over 9 months including 3 weekdays and 1 weekend day each wave 8 (twice during each of the four time windows) Random interval during time windows: 9 a.m.–12 p.m., 12–3 p.m., 3–7 p.m. and 7–10 p.m.	74% completed at least one mEMA survey.	Males were less likely than females to complete mEMAs (*p* < 0.001). Prompts sent in the morning and on the weekend had lower response rates.
Berkman et al., 2014, USA [26]	Signal-contingent	14 days 4 Fixed interval based on individual’s usual mealtimes and bedtime	96% (text messaging group only).	The text group responded at significantly more of the target times than the paper EMA group (paper: M = 70% valid response rate vs. text M = 96%, *p* < 0.001).
Borgogna et al., 2015. [27], Doan et al., 2021, Grenard et al. [28], 2013, USA [29]	Signal-contingent and event-contingent (self-initiated)	7 days (2 school days, 4 non-school days) Random interval—on school days (one between 3 and 6 p.m. and one between 6 and 9 p.m.) and on non-school days, one each in the following 3 h time windows: 9 a.m.–12 p.m., 12 p.m.–3 p.m., 3 p.m.–6 p.m. and 6 p.m.–9 p.m. Evening report 6 p.m.–11.45 p.m. nightly	71% of the random prompts and 95% of the scheduled evening reports.	NR
Cerrada et al., 2016, USA [30]	Signal-contingent and event-contingent (self-initiated)	7 days 5 Random interval within 3 h time windows between the hours of 8 a.m. and 11 p.m.	78%	Less likely to respond to prompts on weekend days relative to weekdays (*p* < 0.001).
Chmurzyńska et al., 2018 [31], Chmurzyńska et al., 2021, Poland [32]	Signal-contingent	7 days (10 days for validation study) 4 Fixed interval: 9 a.m., 1 p.m., 5 p.m. and 9 p.m., 4 h apart	Validation study only: 84% replied to at least three prompts on at least five days.	There was no difference in the response rate between normal weight and overweight/obese individuals.
Comulada et al., 2018 [33], Swendeman et al., 2018, USA [34]	Signal-contingent	6 months 4 Fixed interval every 3 h (selected by each participant ensuring they cover morning, midday and late afternoon time frames). Daily end-of-day prompt	74% of the total number of days over which participants were followed.	mEMA was more likely to be filled out in the morning and decreased as the day went on (all *p* < 0.01). Lower adherence was found on the weekends (*p* < 0.01).
Cummings et al., 2019, USA [35]	Signal-contingent	4 days (2 weekdays and 2 weekend days) 15 Fixed interval: once every hour (excluding sleep hours) from awakening to bedtime	83% of the daily prompts answered.	NR
Finkelstein-Fox et al., 2020, USA [36]	Signal-contingent	11 days Once a day Fixed interval at 8 p.m., 24 h apart	89%	NR
Hofmann et al., 2014, Germany [37]	Signal-contingent	7 days 7 Random interval throughout the 14 h time window which was divided into 7 blocks of 2 h	92% A small fraction (0.3%) of signals was only partially completed. The remaining 7.5% of signals were not responded to at all.	NR
Holmes et al., 2014, Australia [38]	Signal-contingent	7 days 7 Random interval between hours of 10 p.m. and 8 p.m. with a minimum of 1 h between prompts	NR	NR
Inauen et al., 2016, Germany [39]	Signal-contingent	7 days 5 Fixed interval at 11 a.m., 2 p.m., 5 p.m., 8 p.m. and 11 p.m., 3 h apart	91%	NR
Jeffers et al., 2020, USA [40]	Signal-contingent	6 days (Consecutive days from Thursday to Tuesday, covering weekdays and weekends) 6 Fixed interval from 9:30 a.m. to 10 p.m., 2.5 h apart	86% Total mean number of prompts responded to was 30.6 and 5.1 prompts per day.	NR
Konig et al., 2021, Germany [41]	Event-contingent, self-initiated	8 days N/A	NR	NR
Langlet et al., 2020, Sweden [42]	Event-contingent, self-initiated and device-initiated	14 to 21 consecutive days Reported on weekday and weekend days N/A	73% (on average, recorded 2.2 out of 3 main meals/day).	There was an increase in reporting frequency from 70% to 76% of the 3 expected main meals per day from week one to week two.
Laska et al., 2011, USA [43]	Event-contingent, self-initiated	7 days N/A	NR	NR
Lin et al., 2020, Taiwan [44]	Signal-contingent	14 days 4 Random interval during the 6 h time window: 3 a.m.–9 a.m., 9 a.m.–3 p.m., 3 p.m.–9 p.m. and 9 p.m.–3 a.m. Adjusted to individual’s shift work schedule and wake–sleep pattern	57% Mean number of days with mEMA surveys: 12.5. Mean number of completed surveys per person-day: 2.5.	The length of mEMA period (14 days) may have increased burden and resulted in low response rates.
Maher et al., 2020, USA [45]	Signal-contingent	7 consecutive days (covering weekday and weekend days) 5 Random intervals between 9:30 a.m. and 10:30 p.m.	85%	EMA compliance did not differ by time of day, day of week, number of steps taken in the two hours prior to the EMA prompt, sex, or BMI (*p* > 0.05).
McNaughton et al., 2020 [46], Pendergast et al., 2017, Australia [47]	Event-contingent, self-initiated	3 to 4 non-consecutive days including 1 weekend day over a period of 2 weeks N/A	88% of participants completed at least one EMA entry on one allocated recording day in the FoodNow app.	NR
Munasinghe et al., 2020, Australia [48]	Signal-contingent	16-week schedule of follow-up EMAs over 22-weeks Once a day Random interval either between 8 a.m. and 10 a.m. or between 3 p.m. and 8 p.m., 24 h apart	45% completed one or more EMAs, and <1% completed all 96 EMAs over the follow-up period.	NR
Pannicke et al., 2021, Austria [49] and Germany	Signal-contingent	14 consecutive days 4 Fixed interval at 9 a.m., 1 p.m., 5 p.m. and 9 p.m., 4 h apart	85%	NR
Reader et al., 2018, USA [50]	Signal-contingent	7 days 4 Random interval between 7 a.m. and 10 p.m., at least 1.5 h apart	58%	NR
Reichenberger et al., 2018, Austria and Germany [51] Study 1	Signal-contingent	6 days each in a two-week period 5 Fixed interval at 10 a.m., 1 p.m., 4 p.m., 7 p.m. and 10 p.m., 3 h apart	87% for signal-contingent. NR for event-contingent.	NR
Reichenberger et al., 2018, Austria and Germany [51] Study 3	Signal-contingent	7 days 6 Fixed interval at 9 a.m., 11:30 a.m., 2 p.m., 4:30 p.m., 7 p.m. and 9:30 p.m., 2.5 h apart	85% of daily signals.	NR
Reichenberger et al., 2021, Austria and Germany [52] Study 1	Signal- and event-contingent, self-initiated	8 days 6 Fixed interval at 9 a.m., 11:30 a.m., 2 p.m., 4:30 p.m., 7 p.m. and 9:30 p.m., 2.5 h apart	86% of intraday signals.	NR
Reichenberger et al., 2021, Austria and Germany [52] Study 2	Signal-contingent	14 days 4 Fixed interval at 9 a.m., 1 p.m., 5 p.m. and 9 p.m., 4 h apart	85% of intraday signals.	NR
Richard et al., 2017, Austria [53]	Signal-contingent	7 days 5 Fixed intervals at 10 a.m., 1 p.m., 4 p.m., 7 p.m. and 10 p.m., 3 h apart	88% of all possible EMA prompts were answered.	NR
Richard et al., 2019, Austria [54]	Signal-contingent	7 days 5 Fixed interval at 10 a.m., 1 p.m., 4 p.m., 7 p.m. and 10 p.m., 3 h apart	91% of EMA prompts were answered.	NR
Rodgers et al., 2018, Australia [55]	Signal-contingent	7 days (covering weekend/weekdays) 7 Random interval between 10 a.m. to 8 p.m., minimum of 1 h apart	Average number of surveys completed: 41.5.	No associations with age (*p* = 0.99), BMI (*p* = 0.21), intentions (*p* = 0.20), behaviours (*p* = 0.48), education (*p* = 0.75), employment (*p* = 0.26), living status (*p* = 0.70), or relationship status (*p* = 0.92).
Schultchen et al., 2019, Austria and Germany [56]	Signal-contingent	7 days 6 Fixed interval at 9 a.m., 11:30 a.m., 2 p.m., 4:30 p.m., 7 p.m. and 9:30 p.m., 2.5 h apart	On average, 84% of prompted signals were completed.	NR
Schuz et al., 2015 [57], Schuz et al., 2015, Australia [58]	Signal-contingent and event-contingent, self-initiated	10 days Signal-contingent: 3–5 times a day during the waking hours of the day, random interval ∙	90% of snack reports completed.	NR
Serafica et al., 2018, USA [59]	Event-contingent, self-initiated	6 days N/A	Mean compliance of 54% across 6 days.	NR
Seto et al., 2016, China [60]	Signal-contingent	14 days 5 Random interval	Ample compliance.	NR
Spook et al., 2013, Netherlands [9]	Signal-contingent	7 consecutive days 5 Fixed interval at 8 a.m., 12 p.m., 3:30 p.m., 6:30 p.m. and 9:30 p.m., ~3 h apart	Compliance declined 46% over study period, self-reported compliance indicated a smaller decrease in compliance (29%).	No time of day differences between morning, early afternoon, late afternoon, early evening and late evening.
Strahler et al., 2018 [61], Strahler et al., 2020, Germany [62]	Signal-contingent	4 days (Tuesdays to Fridays), weekdays only 5 Fixed interval at 30 min after awakening, 11 a.m., 2 p.m., 6 p.m. and 9 p.m., ~3 h apart	98%	No statistically significant day effect in food/drink consumption (all *p* > 0.06) and subjective states (all *p* > 0.43).
Thomas et al., 2011, USA [63]	Signal-contingent	7 days; given option of up to 10 days depending on return of borrowed device 6 Semi-random interval at 8:30 a.m., 11:10 a.m., 1:50 p.m., 4:30 p.m., 7:10 p.m. and 9:50 p.m., 2 h and 40 min apart	71% of EMA prompts.	No associations with time of day or number of EMA days.
Tomiyama et al., 2009, USA [64]	Signal-contingent	2 days 24 Fixed interval: once an hour (±10 min), 1 h apart	On average, 118 participants completed 2834 diary observations for an average of 24 hourly entries each.	NR
Villinger et al., 2021, Germany [65]	Event-contingent, self-initiated	8 days N/A	NR For high compliance and support data collection, individuals selected their own two reminders (morning and evening)	NR
Wahl et al., 2017, Germany [66]	Event-contingent, self-initiated	8 days N/A	Satisfactory compliance rate (M = 3.4 meals or snacks/day). No selective reporting of certain food items.	NR
Wahl et al., 2020, Germany [67]	Event-contingent, self-initiated	8 days N/A	NR	NR

App = smartphone application, NR = not reported, N/A = not applicable, mERA = mHealth evidence reporting and assessment checklist, mHealth = mobile health, mEMA = mobile-based ecological momentary assessment, PDA = personal digital assistant. Note: Articles reporting on the same data set or sample were merged and presented in this Table as an independent study (*n* = 39).

**Table 4 healthcare-10-01329-t004:** Quality assessment of eligible mobile-based ecological momentary assessment (mEMA) studies (*n* = 39) in young adults using the Joanna Briggs Institute Critical Appraisal Tools for analytical cross-sectional and longitudinal cohort studies [22]. Adapted based on the quality assessment table presented in Shi, Davies and Allman-Farinelli [23].

First Author, Publication Year, Country	Q1	Q2	Q3	Q4	Q5	Q6	Q7	Q8	Q9	Q10	Q11	Quality
Cross-sectional studies												
Berkman et al., 2014, USA [28]	No	Yes	Unclear	N/A	No	N/A	Unclear	Yes *	N/A	N/A	N/A	Poor
Borgogna et al., 2015 [29], Doan et al., 2021 [30], Grenard et al., 2013, USA [31]	Yes	Yes	Yes	N/A	Yes	Yes	Unclear	Yes *	N/A	N/A	N/A	Fair
Cerrada et al., 2016, USA [32]	Yes	Yes	Yes	N/A	Yes	Unclear	Yes	Yes *	N/A	N/A	N/A	Fair
Chmurzynska et al., 2018 [33], Chmurzynska et al., 2021, Poland [34]	Unclear	Yes	Yes	N/A	Yes	Unclear	Yes	Yes *	N/A	N/A	N/A	Fair
Comulada et al., 2018 [35], Swendeman et al., 2018, USA [36]	Yes	Yes	Yes	N/A	Yes	Unclear	Yes	Yes *	N/A	N/A	N/A	Fair
Cummings et al., 2019, USA [37]	Yes	Yes	Yes	N/A	Yes	Yes	Yes	Yes *	N/A	N/A	N/A	Good
Finkelstein-Fox et al., 2020, USA [38]	No	Yes	Yes	N/A	Unclear	Unclear	Yes	Yes *	N/A	N/A	N/A	Poor
Holmes et al., 2014, Australia [40]	No	Yes	Yes	N/A	No	N/A	Yes	Yes *	N/A	N/A	N/A	Fair
Inauen et al., 2016, Germany [41]	Yes	Yes	Yes	N/A	Yes	Unclear	Unclear	Yes *	N/A	N/A	N/A	Fair
Jeffers et al., 2020, USA [42]	Unclear	Unclear	Yes	N/A	Yes	Unclear	Unclear	Yes *	N/A	N/A	N/A	Poor
Konig et al., 2021, Germany [43]	Unclear	Yes	Unclear	N/A	No	N/A	Unclear	Yes *	N/A	N/A	N/A	Poor
Langlet et al., 2020, Sweden [44]	No	Unclear	Unclear	N/A	No	N/A	Unclear	Yes	N/A	N/A	N/A	Poor
Laska et al., 2011, USA [45]	No	Yes	Unclear	N/A	Unclear	Yes	Unclear	Yes *	N/A	N/A	N/A	Poor
Lin et al., 2020, Taiwan [46]	Yes	Yes	Unclear	N/A	Yes	Yes	Yes	Yes *	N/A	N/A	N/A	Fair
Maher et al., 2020, USA [47]	Yes	Yes	Unclear	N/A	Yes	Yes	Unclear	Yes *	N/A	N/A	N/A	Fair
McNaughton et al., 2020 [48], Pendergast et al., 2017, Australia [49]	No	Yes	Yes	N/A	Yes	Unclear	Yes	Yes *	N/A	N/A	N/A	Fair
Pannicke et al., 2021, Austria and Germany [51]	Unclear	Yes	Unclear	N/A	No	N/A	Unclear	Yes *	N/A	N/A	N/A	Poor
Reader et al., 2018, USA [52]	No	Unclear	Yes	N/A	Yes	Yes	Unclear	Yes *	N/A	N/A	N/A	Poor
Reichenberger et al., 2018, Austria and Germany [53] (Study 1)	No	Unclear	Unclear	N/A	Yes	Yes	Unclear	Yes *	N/A	N/A	N/A	Poor
Reichenberger et al., 2018, Austria and Germany [53] (Study 3)	No	Unclear	Unclear	N/A	Yes	Yes	Unclear	Yes *	N/A	N/A	N/A	Poor
Reichenberger et al., 2021, Austria and Germany [54] (Study 1)	Unclear	Yes	Yes	N/A	No	N/A	Yes	Yes *	N/A	N/A	N/A	Fair
Reichenberger et al., 2021, Austria and Germany [54] (Study 2)	Unclear	Yes	Yes	N/A	No	N/A	Yes	Yes *	N/A	N/A	N/A	Fair
Richard et al., 2017, Austria [55]	Unclear	Yes	Yes	N/A	Yes	Yes	Unclear	Yes *	N/A	N/A	N/A	Fair
Richard et al., 2019, Austria [56]	No	Unclear	Unclear	N/A	Yes	Yes	Unclear	Yes *	N/A	N/A	N/A	Poor
Rodgers et al., 2018, Australia [57]	No	Unclear	Unclear	N/A	Yes	Yes	Unclear	Yes *	N/A	N/A	N/A	Poor
Schultchen et al., 2019, Austria and Germany [58]	No	Yes	Yes	N/A	Yes	Yes	Yes	Yes	N/A	N/A	N/A	Fair
Schuz et al., 2015 [59], Schuz et al., 2015, Australia [60]	Yes	Yes	Unclear	N/A	Yes	Yes	Unclear	Yes *	N/A	N/A	N/A	Fair
Serafica et al., 2018, USA [61]	No	Unclear	Unclear	N/A	No	N/A	Unclear	N/A	N/A	N/A	N/A	Poor
Seto et al., 2016, China [62]	No	Unclear	Unclear	N/A	No	N/A	Unclear	Yes *	N/A	N/A	N/A	Poor
Spook et al., 2013, The Netherlands [9]	Unclear	Yes	Unclear	N/A	Unclear	Unclear	Unclear	No	N/A	N/A	N/A	Poor
Strahler et al., 2018 [63], Strahler et al., 2020, Germany [64]	Yes	Yes	Unclear	N/A	Yes	Unclear	Yes	Yes *	N/A	N/A	N/A	Fair
Thomas et al., 2011, USA [65]	Yes	Yes	Yes	N/A	Yes	Yes	Unclear	Yes *	N/A	N/A	N/A	Fair
Tomiyama et al., 2009, USA [66]	No	Unclear	Unclear	N/A	No	N/A	Unclear	Yes *	N/A	N/A	N/A	Poor
Villinger et al., 2021, Germany [67]	No	Yes	Unclear	N/A	Yes	Yes	Unclear	Yes *	N/A	N/A	N/A	Poor
Wahl et al., 2017, Germany [68]	No	Yes	Unclear	N/A	Yes	Yes	Unclear	Yes *	N/A	N/A	N/A	Poor
Wahl et al., 2020, Germany [69]	No	Yes	Yes	N/A	Unclear	Unclear	Unclear	Yes	N/A	N/A	N/A	Poor
Cohort Studies												
Ashurst et al., 2018 [26], Bruening et al., 2016 [27], Bruening et al., 2016, USA [8]	Yes	Yes	Yes	Yes	Unclear	Yes	Yes	No	N/A	N/A	Yes	Fair
Hofmann et al., 2014, Germany [39]	Unclear	Yes	Yes	Yes	Yes	N/A	Unclear	Yes	No	No	Yes	Poor
Munasinghe et al., 2020, Australia [50]	Yes	Yes	Yes	Yes	No	N/A	Yes	Yes	Yes	NR	Yes	Fair

Overall quality rating of each study: poor as 3 or more ‘no’ or ‘unclear’, fair as 1 to 2 ‘no’ or ‘unclear’ and good as only ‘yes’ or ‘not applicable (N/A)’ responses. * A cross-sectional study that used logistic regression/inferential statistics. Caution is needed when interpreting associations. N/A = not applicable; NR = not reported. Question 4 is N/A for all 39 independent studies because this review did not target individuals from a clinical population. Note: Articles reporting on the same data set or sample were merged and presented in this Table as an independent study (*n* = 39).

## Data Availability

Not applicable.

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
