# Peer review of "The Use of Mobile-Based Ecological Momentary Assessment (mEMA) Methodology to Assess Dietary Intake, Food Consumption Behaviours and Context in Young People: A Systematic Review"

_healthcare, 2022, doi:10.3390/healthcare10071329_

Round 1
Reviewer 1 Report
This review followed the Preferred Reporting Items for Systematic Reviews and Meta-Analysis (PRISMA) checklist.The data covered from 1 January 2008 to 8 September 2021.The inclusion and exclusion criteria of the review were developed using a modified PICO framework based on population, EMA measurement method, setting and outcomes.The author used STROBE Checklist for Reporting EMA studies.The references are appropriate and adequate.
Author Response
Thank you for providing us with feedback from three independent referees on our manuscript “The use of mobile-based ecological momentary assessment (mEMA) methodology to assess dietary intake, food consumption behaviours and context in young people: A systematic review”. We are grateful that the reviewers thought the review adhered to correct methodology, was comprehensive and had a detailed discussion.
We trust that the following acceptably addresses the reviewers’ comments and that our manuscript is suitable for publication in Healthcare.
Reviewer 2 Report
This manuscript reviews mEMA (Mobile-based ecological momentary assessment) methodology and its feasibility, acceptability, and validity as a dietary assessment method in young people. The review article aims to the current review aimed to close this knowledge gap by focusing on three key objectives, including examining the methodology of studies using mEMA to measure various aspects of food consumption, evaluating the administration methods of mEMA, and assessing its feasibility, acceptability, and validity as a dietary assessment method.
It reported that mEMA appears to be a feasible and acceptable methodology to assess dietary intake and food consumption in near real-time. I believe the manuscript provides a comprehensive overview and discussion with details. I would suggest accepting the manuscript in the current format.
Author Response

(The authors gave the same response as above.)

Reviewer 3 Report
Thank you very much for allowing me to review the review article entitled “The use of mobile-based ecological momentary assessment (mEMA) methodology to assess dietary intake, food consumption behaviors and context in young people: A systematic review.” (healthcare-1803387).
This study is based on Mobile ecological momentary assessment (mEMA) has the advantage of being incorporated seamlessly into daily living by engaging the individual to provide samples of information in short bursts as it occurs in real time.
This review evaluated Mobile-based ecological momentary assessment (mEMA) methodology and the feasibility, acceptability and validity as a dietary assessment method in young people, on three key objectives: (1) examine the methodology of studies using mEMA to measure various aspects of food consumption, (2) evaluate the administration methods of mEMA, and (3) assess its feasibility, acceptability and validity as a dietary assessment method.
This systematic review followed the Preferred Reporting Items for Systematic Reviews and Meta-Analysis (PRISMA) checklist and the protocol is registered with the Open Science Framework (registration DOI: 10.17605/OSF.IO/WPC7Y;https://osf.io/wpc7y ).They used PICO for eligibility criteria.Five databases were searched from January 2008 to September 2021, at the end 39 independent studies were included demonstrating a wide variation in mEMA methods.
In short, it is a very interesting topic, such as the approach to the situation of the declaration of nutritional intake among young people using new technologies .
The methodology developed is the correct one for a systematic review, it is well written and the discussion is reflexive and its conclusions are derived from the results obtained.
Author Response

(The authors gave the same response as above.)
